# The effect of recall period on reported out-of-pocket health expenditure in Ghana

Isaiah Awintuen Agorinya[1,2,3,4]*, Amanda Ross[1,2], Gabriela Flores[5], James Akazili[4,6,7]*, Tessa Tan-torres Edejer[5], Kim van Wilgenburg[8], Maxwell Ayindenaba Dalaba[3,6], Nathan Kumasenu Mensah[9], Le My Lan[1,2,10], Yadeta Dassie Bacha[11], Jemima Sumboh[6], Abraham Rexford Oduro[6], Fabrizio Tediosi[1,2]

1 Department of Epidemiology and Public Health, Swiss Tropical and Public Health Institute, University of Basel, Basel, Switzerland, 2 Institute of Health Research, University of Health and Allied Sciences, Ho, Ghana, 3 Department of Epidemiology and Biostatistics, Fred N. Binka School of Public Health, University of Health and Allied Sciences, Ho, Ghana, 4 INDEPTH-Network Secretariat, Accra, Ghana, 5 World Health Organization (WHO), Geneva, Switzerland, 6 Navrongo Health Research Centre, Navrongo, Ghana, 7 Department of Population, Family and Reproductive Health, School of Public Health, CK Tedam University of Technology and Applied Sciences, Navrongo, Ghana, 8 Department of Public Health and Management Health Economics, Erasmus University, Rotterdam, the Netherlands, 9 Department of Health Information Management, University of Cape Coast, Cape Coast, Ghana, 10 FilaBavi Health and Demographic Surveillance Site, Hanoi, Vietnam, 11 Department of Public Health, College of Health and Medical Sciences, Haramaya University, Harar, Ethiopia

* iagorinya@gmail.com (IAA); jakazili@gmail.com (JA)

## Abstract

### Background

Out-of-pocket health payments (OOPs) are a key indicator of health financing systems' performance. Measuring OOPs through household surveys is challenging and yet it is the primary source of information in the absence of comprehensive data on user charges in the public sector and market data from the private sector. The choice of the recall period has been identified as a source of bias in previous studies. This study investigates the effect of two different types of recall periods on the agreement between OOPs reported by households and providers.

### Methods

Households were sampled for our community survey from the Navrongo Health and Demographic Surveillance System, Ghana. Two versions of a health expenditure module were developed differing only in the recall periods, "shorter recall periods" 2 weeks for medicines and outpatient care, 3 months for preventive care and 6 months for inpatient care and medical products. The longer recall periods were 4 weeks, 6 months and 12 months. Households from both community and provider sampling were randomly assigned to the two questionnaires. The providers included the hospital, one clinic and, health facilities and drug shops in the area. We estimated the ratio between the overall mean household OOPs and overall mean provider OOPs.

**Data availability statement:** All relevant data are within the manuscript and its Supporting information files.

**Funding:** This project was funded by the INDEPTH-Network in Accra (http://www. indepth-network.org/) through a grand from Bill and Melinda gates foundation, grant number OPP1113162. GF, TE, KvW, AR were partially supported by WHO. The funders had no role in study design, data collection and analysis, decision to publish, or preparation of the manuscript.

**Competing interests:** The authors have declared that no competing interests exist.

**Abbreviations:** CAPI, Computer Assisted Personal Interviews; COICOP, Classification of Individual Consumption according to purpose; DHS, Demographic and Health Survey; GLSS, Ghana Living Standards Survey; HHS, Household Health Survey; HIC, High income countries; iHOPE, INDEPTH-Network Household Out-of-pocket Expenditure; LMIC, Low and Middle-income Countries; LSMS, Living Standards Measurement Survey; NHRC, Navrongo Health Research Centre; NHDSS, Navrongo Health and Demographic Surveillance System; NHA, National Health Accounts; NHIS, National Health Insurance Scheme; OOPs, Out-of-Pocket Health Spending; OPD, Out-patient Department; SHA, System of Health Accounts; WHO, World Health Organization; WHS, World Health Survey.

We assessed agreement between the individual matched household-provider OOPs using Bland-Altman analysis.

## Findings

The short and long-recall period versions of the questionnaires were administered to 746 and 480 households with matching success to provider records of 72% and 84%, respectively. The most common spending categories were inpatient care and medicines in this sample. The overall mean OOPs reported by the households were higher than provider records for both recall periods. For matched household-provider data, there was no evidence of a difference in the agreement between the household and provider OOPs for inpatient care, the ratio of household to provider for the 12 months recall was estimated to be 0.74 (95% CI 0.45, 1.19; p = 0.22) that of the ratio of household to provider for the 6-month period, where less than 1 would indicate better agreement. For medicines, the ratio of 4 weeks to 2 weeks was 1.26 (0.93, 1.39; p = 0.39).

## Conclusion

There were considerable challenges in using provider data to assess the accuracy of reported OOPs in this setting. There was no evidence from this study that the agreement between household and provider data differed by recall period, however the confidence intervals of the effect were wide, and an effect cannot be ruled out. These findings underscore the need for caution in relying solely on provider-reported data and highlight the importance of considering alternative data collection methods to improve the accuracy of healthcare cost assessments.

## Background

Out-of-pocket (OOP) health payments are defined as direct payments individuals make at the point of service to access healthcare which can be in the form of informal payments, user chargers, coinsurance, co-payments or deductibles [1]. These OOPs exclude any prepayment for health services in the form of insurance premiums, government subsidies and or taxes [2].

Globally, WHO estimated that OOP(s) accounted for 44% of current health expenditure in 2019, the latest year for which the evidence is currently available [3]. OOP is used in all countries at all income levels to fund the health system but the extent to which they contribute to it varies markedly by country income group levels. The proportion contributed ranged between 44% in most low-income countries, 40% in low-middle-income countries (LMICs), 34% in upper-middle-middle income countries to 21% in high income countries (HICs) [4–6]. In Ghana, 36% of health care expenditure was OOP in 2019 [3,7].

At the individual level, for some people, out-of-pocket health payments represent a financial barrier to access leading to foregone care. For those making such

payments, on the other hand, they may not prevent themselves from seeking care, but they can be a source of financial hardship. Financial protection in health aims to eliminate both negative outcomes. It is a key component of Universal Health Coverage (UHC) [8,9]. Information on OOPs is used both for evidence-based health financing policy discussions and to track progress towards financial protection in health [10,11].

Household surveys are important for measuring both OOPs and the households' ability to pay in absolute or relative terms to calculate financial protection. OOPs tracked in household surveys are the primary source of information in LMICs to determine their contribution to the overall health spending landscape. Household surveys are the only source of information available across all countries at all income levels to gather information on both household's OOP and their ability to pay.

Despite the importance of household surveys, the design of the module(s) used to collect data on OOP is not standardized, neither across countries, nor within countries over time. One reason for this is the existence of several challenges in gathering such information. The most common ones are the Living Standards and Measurement Surveys (LSMS), the Household Budget Surveys (HBS), the Socio-Economic Surveys (SES) and Income and Expenditure Surveys (IES), as well as household expenditure and utilization surveys [6–8]. These surveys differ in the level of comprehensiveness and specificity of the health expenditure questions; the module used to collect the information on health spending, the overall focus of the survey and the recall period. The latter is the focus of this paper.

Differences in recall periods contribute to recall bias problems [5,9,12–14]. According to nationally representative survey-based studies, a short recall period leads to a larger annualized estimate of OOP than a long recall period in most countries [5,7,15]. However, very few studies have investigated the impact of recall period on reported OOPs tracked in household surveys [5,7,11].

What the optimum recall period should be is not well established. Stull *et al*. found that a single recall period is not appropriate for measuring and understanding all outcomes [16]. In the case of health payments, a single recall period is unlikely to be relevant given their different frequencies and costs. When different recall periods are used for health expenditures, the common choices are one month, six months or 12 months but there is no standard.

It is therefore important to investigate the effect of the recall period on the accuracy and reliability of data collected in household surveys on the components of out-of-pocket health expenditure [7]. To assess accuracy, reported OOPs need to be compared to actual cost although these are difficult to measure reliably. Health facility/provider records are an objective measure which does not depend on participants' recall but does depend on capturing information from all the relevant health facilities. This study investigates the effect of different recall periods on amounts spent on various health goods and services out-of-pocket by comparing the agreement between household respondents in the community and health facility records in Navrongo, Ghana as part of the INDEPTH-network household out of pocket expenditure (iHOPE) project.

## Methods

### Study setting

This study was implemented at the Navrongo Health and Demographic Surveillance System (NHDSS) site located in the northern part of Ghana. The site includes two administrative districts with an estimated total population of 160,000. Within this site, there is one hospital, a health research institution, one private clinic, seven health centers, and 27 community-based health compounds. A number of pharmacies and licensed chemical shops, petty traders, drug peddlers, herbalists, faith-based and traditional healers also operate in the area. The NHDSS maintains a demographic surveillance system that routinely collects demographic, health and and economic data of households within its operational areas [17].

### Health care financing in the study setting

Ghana is one of very few countries to have enacted a legislation (National Health Insurance Act 2003 (Act 650) and begun the transition to universal health insurance coverage (National Health Insurance Scheme, NHIS) to replace the OOPs

previously referred to as "Cash and Carry" system. The financing scheme is generally progressive and is largely financed through tax (Akazili, 2011) and a small proportion from contributions and donations. In 2014, the scheme covered only 40% of Ghana's population (10.5 million active subscribers) with 69% of these exempted from any form of payment to the scheme (Wang, Otoo & Dsane-Selby, 2017). The exempted group include indigent people, pregnant women and very poor households covered by the social intervention programme called "Livelihood Empowerment Against Poverty" (LEAP). The National Health Insurance Scheme covers 95% of disease conditions reported in Ghana with services including primary curative care to care at tertiary facilities.

Despite the existence of the NHIS in Ghana, out-of-pocket payments for health care persist within the health system contributing to 48% 2005/2006 to 36% by 2019 of the health care financing in Ghana. All subscribers accessing health care from NHIS accredited health facilities are assured of free services but maybe exposed to spending out-of-pocket for medicines, laboratory tests, vaccinations and other consumables which may not be available at the provider due for example to stock-outs (Addae-korankye, 2013). The uninsured population (accounting for about 60% in 2014) will be required to pay out-of-pocket to be able to access health care (Wang, Otoo & Dsane-Selby, 2017). Therefore, we expect some level of OOPs within the Ghana health system especially for medicines, preventive care, and hospitalization. OOPs in Ghana are regressive and consequently, diminish the overall level of progressivity in the health care funding in Ghana (Akazili, Gyapong& McIntyre, 2011). "Under the table payments" or informal payments for health care were not found to be practicedin the study area. This was established during the pilot study phase of the iHOPE project which was conducted four weeks prior to the start of the actual data collection.

## Study design

The agreement between household and provider OOPs was compared for two different questionnaire versions using different recall periods (Table 1). Households were randomized into two groups. First, new modules of health expenditure questions were designed and integrated into existing survey tools (Ghana Living Standards Survey 6 questionnaire), then cross-sectional household and provider data collection was carried out in the field using the new questionnaires and then the survey data was matched with the provider data.

## Study population and sampling

The study population constituted all households registered in the Navrongo Health Research Demographic Surveillance System (NHDSS). Two sampling strategies were employed: standard household-based sampling was adopted for outpatient, medicines, and preventive care whilst provider-based inpatient sampling was adopted for inpatient care expenditures due to the low frequency of inpatient spending. The sample size was based on the precision of estimating the agreement between household and provider records. As a rule of thumb for the Bland-Altman method of assessing agreement, between 100–200 observations would provide a sample size with sufficient precision of the estimates when assessing

**Table 1. Spending categories and corresponding recall periods.**

| Health spending category | Questionnaire Version1<br>shorter recall period | Questionnaire Version 2<br>Longer recall period |
|---|---|---|
| Inpatient care | 6 months | 12 months |
| Preventive care | 3 months | 6 months |
| Other health services | 2 weeks | 4 weeks |
| Outpatient | 2 weeks | 4 weeks |
| Medicines | 2 weeks | 4 weeks |
| Health products | 6 months | 12 months |

agreement [18]. Sample sizes for the survey were computed to achieve100 household with positive OOPs per spending category. However, we accepted 50 observations for each spending category per questionnaire version as adequate in spending categories where is was not feasible to obtain the 100 matched household observations.

**Household sample.** The probability of spending on outpatient care was 15.5% within a two-week period [*unpublished Navrongo DHMT, 2015 data*]. To obtain a sample size of 100 households with outpatient spending in the two-week recall period, the number of households in this group would be 600. We added 10% to account for non-response to arrive at the total sample of 660 households for this questionnaire version. For the four- week recall period, the sample size required to get a minimum of 100 households who incurred health expenditure would be 400, adding 10% to account for non-response gives a target of 440 households.

**Health provider (Facility) sample.** Only one hospital (public provider) in the study area provides inpatient care services. From this provider database, we randomly selected 220 households with positive expenditure to form the sample with inpatient care. Each recall period group (6 months and 12 months) was randomly assigned to110 households.

**Sampling.** For the household sampling, households were sampled using the Navrongo Health and Demographic Surveillance System (NHDSS). The Navrongo DSS is divided into five zones (North, South, East, West and Central), sub-zones and clusters. The households were selected randomly from the DSS database which contains all 33,000 households. The other part of the sample was obtained from health providers as described above in the health provider sampling section. The flowchart summarizes the target and actual sample for each questionnaire version (Fig 1).

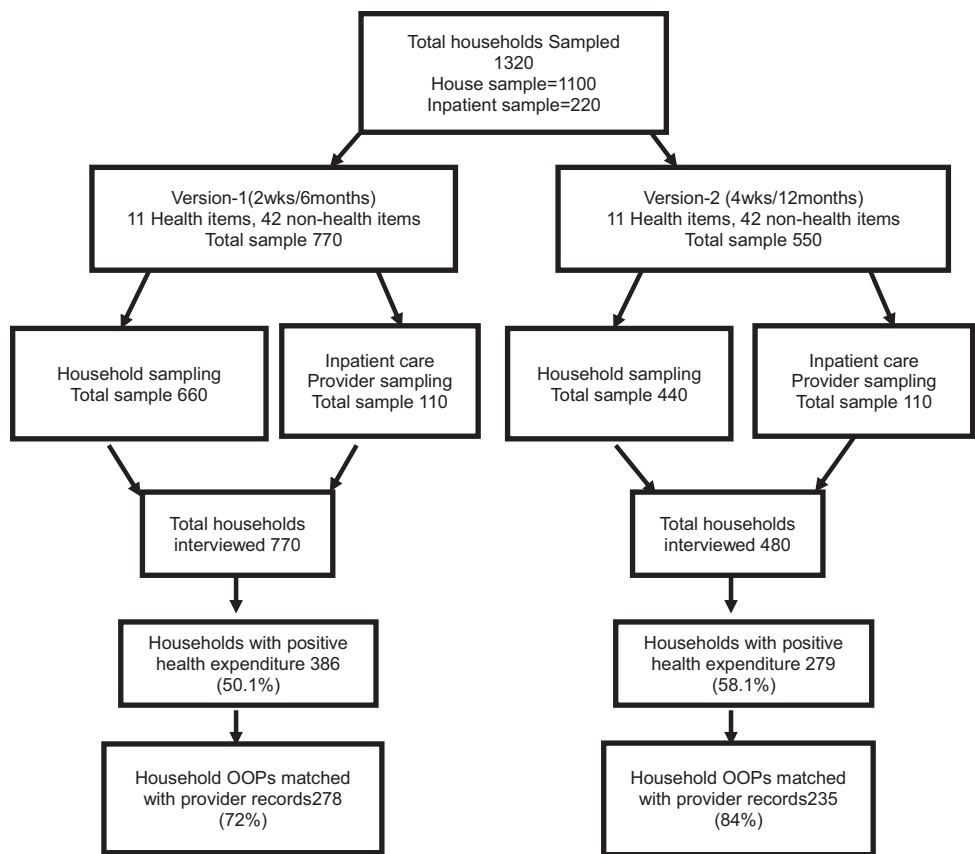

**Fig 1. Flowchart for the target and actual sample.**

**The health provider (Facilities).** The health providers included all public and private health care providers operating within the study area. They include one hospital, one clinic, seven health centers and ten high volume pharmacy shops and around 50 chemical shops. This allows us to investigate agreement for the main spending categories reported by households. To obtain data from the providers, we identified and selected only providers that kept transactional records or were capable of recording such information and placed field staff to assist in recording transactions in providers that did not previously keep records.

**Data collection instruments.** *Household data collection instrument:* A health expenditure and utilization household survey was developed by WHO drawing on the structure of the World Health Survey [19] and adapted to the Ghana Living Standards Survey 6 (GLSS6) [20]. The structure of the survey instrument included a household level questionnaire with questions about household OOPs as part of an expenditure module asked to a single respondent within the household, and an individual level questionnaire with information on utilization and health expenditures answered by the same respondent. The focus of this study is on the household level questionnaire. From the household questionnaire, 11 questions on OOPs were included in the survey. The questions were developed to map to the UN statistical classification of individual consumption according to purpose COICOP-2018. The final structure of this household questionnaire is illustrated S1 Fig. S2 Table also shows how the health expenditure questions were framed and gives the instructions on how the questions were administered.

The respondent for the expenditure module was the head of the household or any other knowledgeable person assigned by the household head to provide such information. Trained field workers conducted face-to-face interviews using computer assisted personal interviews (CAPI). The questionnaire was piloted among households who were not part of the study sample. During the piloting, "under the table" or informal payments and levels of OOPs for different spending categories were enquired about. The pilot lasted two weeks (May 2017) and allowed the study investigators to fine-tune the questionnaires and the research design before final data collection. Data collection lasted 4 months, that is from July 2017 to October 2017 after the required sample size was achieved. Written informed consent was obtained from every household head before the study questionnaire was administered.

*Provider (facility) data collection and matching*: A template (S3 Table) was developed to collect patient data from different types of health care providers (all pharmacy and licensed chemical shops) who did not have previous experience in collecting patient data. The template was used to collect the name, address, phone number, referral status, reason for consultation and cost of treatment/service. This information was requested from patients at the point of paying for the services after they had consented to be part of the study. Two of the high-volume pharmacy shops requested and received additional staff to assist in recording patient data. Public providers already have experience collecting patient data. OOPs records were extracted from their records database or books by the project field team. All provider records were collected for a total of 12 months to cover the different recall periods. Hospital records covering a period of 12 months were extracted to capture inpatient expenditures over the past 12 and 6 months.

Every household that reported OOPs within a given recall period for any of the spending categories was asked for additional details about the transaction(s) and the provider(s) with whom transactions occurred. The details facilitated the matching process. Matching of household OOPs to provider records was done at the individual level but OOPs across household members were aggregated to perform the household-level analysis. S4 Fig shows a flowchart detailing the matching procedure.

## Data analysis

The design of this study makes it possible to estimate the effect of different recall periods on OOP estimates. Two approaches were used. We first compared the overall means of the households OOPs in the two recall period groups. The effect of the different recall periods was estimated as the ratio of the mean OOPs. This is what is typically investigated in published studies. The limitation is that it is not possible to know which recall period leads to the more accurate estimates.

The second approach uses the matched responses to consider the level of agreement between the household responses and the provider data. The matched households are a subset of all households: we tabulated the characteristics of both to compare their characteristics and identify any potential source of bias in the type of households that matched. We then applied the Bland-Altman approach for method comparison [21,22].For each spending category and questionnaire version, we estimated the overall agreement between the household and provider OOPs and the variability in the agreement between records. We calculated the ratio of households to provider OOPs rather than the difference since the difference was heavily dependent on whether the provider amounts were large or small. We also applied a log-transformation to the ratio before the Bland-Altman analysis as recommended when the distribution is skewed [21,22]. When back transformed to the OOPs scale, this gives us the geometric mean ratio. We present the estimates of variability as 95% limits of agreement which represent the range in which we expect 95% of the observed individual household to provider-ratios to lie.

We then investigated whether recall period affected the agreement between household and provider OOPs by following the regression method of Bland and Altman [21]. To investigate the effect of the questionnaire version, we fitted a regression model with the difference of the log OOPs between household and provider expenditures as the outcome variable [22] and questionnaire version as an explanatory variable. This allows us to estimate the effect of the questionnaire version on the geometric mean ratio of household to provider OOPs. We included a random effect parameter to account for the clustering of the households within clusters defined by the Navrongo DSS [17]. We estimated the effect of the questionnaire version on the variability by regressing the questionnaire version on the absolute values on the residuals of the previous model. Data was analyzed using STATA Version 14, Stata Corp.

### Ethics approval and consent to participate

The Ethical Review Board of the Navrongo Health Research Centre, Ghana (NHRCIRB217) approved for the conduct of the study. Written Informed consent was obtained from all study participants.

### Results

Out of the 1320 households selected, 1,226 (92.9%) were interviewed. Of the households interviewed, 386 (50%) and 279 (58%) reported OOPs expenditure in version 1 and version 2 respectively. For those reporting expenditure, 278 (72%) and 226 (81%) of reported OOPs were successfully matched with their respective provider data at individual levels in version 1 and 2 respectively (S4 Fig). The most frequently reported spending category was medicines in the community-based household sampling and inpatient care for the provider inpatient sample (Table 2). Due to challenges in identifying and locating households sampled from the provider records, only 17% and 41% of the targeted provider sample size was achieved in the 6 month and 12 month recall periods respectively. Consequently version 2 has more households reporting OOPs for inpatient care than

Table 2. Composition of the combined sample of households from the community and provider sampling.

| Spending category | | | | Version-1 Short recall period | Version-2 Long recall period | | | |
|---|---|---|---|---|---|---|---|---|
| | Recall period | Household sampling | Provider sampling | Total households | Recall period | Household sampling | Provider sampling | Total households |
| inpatient care services | 6 months | 89 | 19 | 108 | 12 months | 55 | 45 | 100 |
| Preventive services | 3 months | 18 | 10 | 28 | 6 months | 19 | 2 | 21 |
| Other health services | 2 weeks | 0 | 0 | 0 | 4 weeks | 1 | 0 | 1 |
| Outpatient | 2 weeks | 25 | 8 | 33 | 4 weeks | 10 | 5 | 15 |
| Medicines | 2 weeks | 278 | 17 | 295 | 4 weeks | 185 | 19 | 204 |
| health products | 6 months | 5 | 4 | 9 | 12 months | 1 | 1 | 2 |
| Total | | 415 | 58 | 473 | | 271 | 72 | 343 |

version 1. The results presented in this paper are based on the combined sample including both the community-based household sample and provider-based inpatient sampling. However, results based on the analysis of the data from the household sample alone (which constitute about 90% of the combined sample size) are very similar and are contained in S5 Table.

## Demographic characteristics in all households in the survey and matched households only

The demographic characteristics of the household heads were similar across the two questionnaire versions for both the full combined sample and the matched households only (Table 3). Overall, roughly 65% of the households are headed by males. Only 10% of heads were under 35 years and 58% of household heads were married.

## Proportion of households with health care utilization and expenditure

In the combined sample including both the community and inpatient samples, the proportions of households reporting OOPs in medicines and inpatient care were observed to be higher for the longer compared to the shorter recall period. The

Table 3.  General household and demographic characteristics by questionnaire version.

| | all households | | | | | | matched households | | | | | |
| --- | --- | --- | --- | --- | --- | --- | --- | --- | --- | --- | --- | --- |
| | Version 1 (2wks/6months) | | Version 2 (4wks/12months) | | Total | | Questionnaire Version 1 (2wks/6months) | | Questionnaire Version 2 (4wks/12months) | | Total | |
| Total numberofhouseholds | N=800 n | % | N=480 n | % | % | | N=278 n | % | N=235 n | % | N=513 n | % |
| **Household Head** | | | | | | | | | | | | |
| **Sex** | | | | | | | | | | | | |
| Male | 492 | 61 | 287 | 60 | 61 | | 172 | 62 | 164 | 70 | 336 | 65 |
| **Marital status** | | | | | | | | | | | | |
| Married | 446 | 56 | 272 | 57 | 56 | | 161 | 58 | 141 | 60 | 303 | 59 |
| **Level of Education** | | | | | | | | | | | | |
| No education | 538 | 67 | 289 | 60 | 65 | | 187 | 67 | 140 | 60 | 326 | 64 |
| Primary | 111 | 14 | 104 | 22 | 17 | | 46 | 17 | 53 | 23 | 99 | 19 |
| Junior high school | 70 | 9 | 43 | 9 | 9 | | 23 | 8 | 25 | 11 | 48 | 9 |
| Senior high school | 22 | 3 | 25 | 5 | 4 | | 4 | 1 | 9 | 4 | 13 | 3 |
| Vocational/Technical/College/Graduate | 59 | 7 | 19 | 4 | 6 | | 18 | 7 | 7 | 3 | 27 | 5 |
| **Religion** | | | | | | | | | | | | |
| Christians | 349 | 44 | 236 | 49 | 46 | | 110 | 40 | 118 | 50 | 228 | 44 |
| Islam | 19 | 2 | 35 | 7 | 4 | | 6 | 2 | 10 | 4 | 16 | 3 |
| Traditional | 332 | 42 | 181 | 38 | 40 | | 125 | 45 | 91 | 39 | 216 | 42 |
| No religion | 100 | 13 | 28 | 6 | 10 | | 37 | 13 | 16 | 7 | 53 | 10 |
| **Age group** | | | | | | | | | | | | |
| 15 - 19 | 33 | 4 | 16 | 3 | 4 | | 11 | 4 | 12 | 5 | 23 | 4 |
| 20-34 | 47 | 6 | 34 | 7 | 6 | | 17 | 6 | 18 | 8 | 35 | 7 |
| 35 - 64 | 420 | 53 | 282 | 59 | 55 | | 150 | 54 | 138 | 58 | 288 | 56 |
| 65 + | 300 | 38 | 148 | 31 | 35 | | 100 | 36 | 67 | 29 | 167 | 33 |
| Mean age (SD) | 59 (17) | | 55 (17) | | | | 57 (17) | | 54 (17) | | | |
| **Household size** | | | | | | | | | | | | |
| 1 person | 57 | 7 | 38 | 8 | 7 | | 17 | 7 | 9 | 4 | 26 | 5 |
| 2-5 persons | 421 | 53 | 301 | 63 | 56 | | 127 | 46 | 143 | 61 | 270 | 53 |
| 6 and above | 322 | 40 | 141 | 29 | 36 | | 134 | 48 | 83 | 35 | 217 | 42 |

higher proportion observed for inpatient care in the longer recall period is largely attributed to the disproportionate contribution of samples from the provider inpatient sample into the two-recall period groups as observed in Table 2 (19 households in version 1 compared to 45 households in versions 2). Except for inpatient care, the addition of the provider sample did not influence the distribution of reported expenditures by recall period in the other spending categories (Table 4).

## Comparison of mean household reported OOPs by recall period

The mean of the household OOPs by recall period for the combined community and provider inpatient samples was estimated (Table 5). The variability between households in reported OOPs was large and there was no evidence of any significant differences in mean OOPs between the shorter and longer recall periods for each of the separate spending categories. Compared to the longer recall period, the shorter period tended to produce estimates in the direction of being larger. The medicines category had the greatest number of observations and the mean annual OOP was estimated to be 1.59 (0.88, 2.29) times higher for the shorter compared to the longer recall period.

Table 4. Households reporting out of pocket payments by spending category for all and matched households.

| | Questionnaire Version 1 (Short recall period) | | | Questionnaire Version 2 (Long recall period) | | |
|---|---|---|---|---|---|---|
| | all households | | matched only | all households | | matched only |
| Spending category | Recall period | N = 780 n (%) | N = 278 n (%) | Recall period | N = 480 n (%) | N = 235 n (%) |
| inpatient care services | 6 months | 108 (14) | 35 (12) | 12 months | 100 (21) | 64 (27) |
| preventive services | 3 months | 28 (3) | 20 (7) | 6 months | 21 (4) | 18 (8) |
| Other health services | 2 weeks | 0 (0.0) | 0 (0.0) | 4 weeks | 1 (0.2) | 0 (0.0) |
| Outpatient | 2 weeks | 33 (4) | 20 (7) | 4 weeks | 15 (3) | 11 (5) |
| Medicines | 2 weeks | 295 (38) | 234 (85) | 4 weeks | 204 (43) | 167 (72) |
| health products | 6 months | 9 (1) | 0 (0.0) | 12 months | 2 (0.4) | 1 (0.43) |

Table 5. Comparison of mean OOPs in households by recall period.

| | Questionnaire Version 1 (short recall period) | | | Questionnaire Version 2 (long recall period) | | | Non-annualized ratios | Annualized ratios |
|---|---|---|---|---|---|---|---|---|
| Spending category | N | Household (HH) Mean(SD) | Annualized total Estimates Mean(SD) | N | Household (HH) Mean (SD) | Annualized total Estimates Mean(SD) | Estimated ratio (HH-v1/HH-v2 95% CI | Estimated ratio (HH-v1/HH-v2 95% CI |
| Inpatient | 108 | 462 (1573) | 923 (3145) | 100 | 419 (675) | 419 (675) | 1.10 (0.29, 1.89) | 2.20 (0.53, 3.87) |
| Medicines | 295 | 15 (43) | 358 (1040) | 204 | 19 (38) | 226 (460) | 0.79 (0.44, 1.13) | 1.59 (0.88, 2.29) |
| Outpatient | 28 | 43 (79) | 1027 (1894) | 11 | 27 (23) | 327 (273) | 1.59 (0, 3.22) | 3.14 (0, 6.78) |
| Preventive care | 22 | 25 (29) | 99 (115) | 21 | 93 (241) | 187 (482) | 0.26 (0, 0.82) | 0.53 (0, 1.66) |
| Other medical services | 0 | 0 (0) | 0 (-) | 1 | 200 (-) | 2400 (-) | – | – |
| Health products | 5 | 21 (21) | 21 (21) | 2 | 7 (4) | 7 (4) | 3 (0, 6.77) | 3 (0, 6.77) |
| Annualizedtotal household OOPs | | | 627 (2095) | | | 355 (681) | | 1.79 (1.10, 2.49) |

Note: the currency used is the Ghana cedi (GHc). US$1was equivalent to Ghc4.2 at the time of collecting data. **Short recall period**: 2 weeks outpatient/medicines/other health services, 3 months for preventive care and 6 months for inpatient/medical products. **Longer recall period**: 4 weeks outpatient/medicines/other health services, 6 months for preventive care and 12 months for inpatient/medical products.

We assume OOPs do not vary seasonally. The annualized estimates are based on this assumption to allow for comparison across the recall period annually.

## Mean OOPs reported by households compared with provider data (matched data only)

Household reported health expenditures tended to be higher on average than corresponding provider recorded expenditures, and this was observed in both shorter and longer recall period groups and in all spending categories. However, the difference in OOPs reached statistical significance only for inpatient care and medicines. Expenditure records from health care providers were on average a third of the amount households would report to incur for inpatient care and about half the amount in medicines regardless of the recall period used (Table 6).

## Comparing agreement between individual matched household and provider data by recall period

This part of the analysis focuses on matched OOPs estimates for only transactions for inpatient care and medicine as only a few households reported expenditures on the other spending categories and therefore the sample size did not allow for the Bland-Altman approach.

**Table 6. Mean OOPs household survey and provider OOPs for households that matched with provider data.**

| Spending category | | Questionnaire Version 1 | | | | Questionnaire Version 2 | | |
| | | (short recall period) | | | | (Long recall period) | | |
| | N | Provider OOPs | Household OOPs | Estimated ratio (HH/provider) of the means (95% CI) | N | Provider OOPs | Household OOPs | Estimated ratio (HH/provider) of the means (95% CI) |
| | | Mean (SD) | Mean (SD) | | | Mean (SD) | Mean (SD) | |
|---|---|---|---|---|---|---|---|---|
| Inpatient | 35 | 94 (114) | 298 (322) | 3.17 (1.70, 4.65) | 64 | 144 (167.) | 427 (539) | 2.94 (1.82, 4.10) |
| Medicines | 234 | 5 (5) | 10 (15) | 2.1 (1.66, 2.46) | 167 | 7 (7) | 15 (31) | 2.26 (1.60, 2.91) |
| Outpatient | 11 | 3 (5) | 46 (88) | 14 (0, 38.16) | 11 | 9 (9) | 23 (20) | 2.72 (0.58, 4.87) |
| Preventive care | 9 | 6 (13) | 18 (16) | 2.98 (0, 7.78) | 15 | 21 (60) | 42 (76) | 1.62 (0.56, 2.68) |
| Other medical services | 0 | 0 (0) | 0 (0) | – | 0 | 0(0) | 0 (0) | – |
| Health products | 0 | 0 (0) | 0 (0) | – | 3 | 3 (0.5) | 6 (4) | – |

Note: the currency used is the Ghana cedi (GHC). US$1 was equivalent to GHc4.2 at the time of collecting data. **Short recall period**: 2 weeks outpatient/medicines/other health services, 3 months for preventive care and 6 months for inpatient/medical products. **Longer recall period**: 4 weeks outpatient/medicines/other health services, 6 months for preventive care and 12 months for inpatient/medical products.

**Table 7. Mean bias and variability in measurement of OOPs by recall period.**

| Spending category | Number of observations | Geometric mean of the individual HH:provider ratio | 95% limits of agreement | Estimated effect of the recall period on mean ratio: the ratio of the mean ratios (qu2 vs qu1) & CI & p-value | Estimated effect of recall period on variability: the ratio of standard deviations (qu2 vs qu1 (& CI & p-value |
|---|---|---|---|---|---|
| **Inpatient care** | | | | | |
| 6 month recall period (qu1) | 31 | 2.48 | 0.35–18.2 | – | – |
| 12 months recall period (qu2) | 63 | 1.77 | 0.19–16.5 | 0.74 (0.45 - 1.19) 0.22 | 1.02 (0.77–1.37) 0.87 |
| **Medicines** | | | | | |
| 2 week recall period (qu1) | 235 | 1.37 | 0.40–4.64 | – | – |
| 4 week recall period (qu2) | 169 | 1.42 | 0.38–5.47 | 1.26 (0.93–1.39) 0.09 | 1.24 (1.03–1.49) 0.02 |

Note: Limits of agreement refer to the range in which 95% of the mean ratios are expected to lie. The mean ratio is the mean of the ratios between household OOPs and provider OOPs.

The geometric mean of the individual household to provider ratios was greater than one indicating higher household compared to provider OOPs for both recall periods (Table 7, column 3). This is consistent with the previous analysis using the aggregated means rather than the individual matched household records. There was substantial variation in these individual household-level ratios, shown by the 95% limits of agreement (Table 7, column 4).

We compared the agreement between individual household and provider records by recall period. There was no evidence of a difference in the geometric mean ratios (Table 7, column 5) by recall period for either the medicines or inpatient categories. For inpatient spending, the mean ratio for the 12 month recall was estimated to be 0.74 (0.45, 1.19) times that of the 6 month recall (p = 0.22), and for medicines 1.26 (0.93, 1.39, p = 0.09). The confidence intervals are wide and do not rule out the possibility of an effect of recall period. There was an indication of a small increase in variability for medicines only for the four-week compared to two-week recall period but this was not found for inpatient spending (Table 7, column 6).

## Discussion

This study investigated the effect of recall periods using household health expenditure modules and provider records. In this study, the two major sources of household OOPs were inpatient care and medicines.

The first finding was that shorter recall periods tended to produce higher annualized OOP estimates than longer recall periods in the full sample, although not statistically significant. This is consistent with previous reports of higher estimates for shorter recall periods from a study of hospitalization cost in 43 countries(Lu et al, 2009), a study of health expenditure in Nepal [5] and a study of the share of household expenditure on health [9]. The second finding was that household-reported OOP tended to be higher than the provider records overall in the matched sample. The reasons for this are not known but may stem from poor recall from the household members, problems with provider records and selection of providers, or issues with matching the two together. For these reasons, the provider data is not considered as a validation of the household-reported OOPs except for instances where there is proper documentation of provider data that can serve as verifiable data for validation. This challenge potentially reinforces the challenges in achieving universal health coverage in resource poor countries.

The sample size for the estimation of agreement was affected by number of households that could be successfully matched to their corresponding provider data. Most of the providers had challenges recording and extracting health expenditure records of clients since this was not routinely done. This affected the completeness of the provider data and therefore households with zero expenditures and those without accurate personal details could not be included and consequently affected the final sample size for the analysis. Details of these challenges and how they can be addressed in future studies have been explored by Agorinya et al, 2021 [23].

The third finding, and the main question of the study, was that there was no evidence of a difference in the level of agreement between household and provider reported OOPs for the two sets of recall periods. However, the confidence intervals around the point estimate of effect are wide and an effect of recall period on recall bias cannot be ruled out. We recognize that we could not assess the impact of recall period on the likelihood of remembering a transaction, only on the household and provider amounts for transactions which had been recalled. Other studies on accuracy have had mixed results. A vital and health statistics report [24] found that reporting accuracy for inpatient care decreased significantly after eight months, however Nester and colleagues [25] found no such evidence in their study using bounded and unbounded interviews.

Hijinks and his colleagues [7] also found from 90 surveys in 64 countries using International Household Survey Network (IHSN) that most surveys preferred longer recall (12 months) periods in hospital spending and short recall periods (2 weeks) for outpatient and medicine spending in half of the surveys they evaluated. Several other studies have also confirmed the preference of longer recall period for infrequent events and shorter recall period for frequent events [7,26,27].

Despite these limitations and challenges, this study adds to the body of evidence for guidance on the comparability of health expenditures across different surveys using different recall periods. It also provides information on the feasibility of using provider health records in a rural setting.

## Conclusion

The use of provider data to validate household-reported OOPs presented substantial challenges in this setting, mainly due to issues of completeness. In addition, the study found no discernible effect of the recall period on the alignment between household and provider-reported OOPs. However, the lack of definitive evidence on differences in alignment by recall period, coupled with the wide confidence intervals, underscores the uncertainty surrounding this relationship. As a result, further research is needed to develop robust methods for accurately measure out of pocket health expenditure in similar settings.

## Supporting information

**S1 Fig. Design of Household Health Survey Instrument.**
(DOCX)

**S2 Table. Supplementary material 2.**
(DOCX)

**S3 Table. Supplementary material 3.**
(DOCX)

**S4 Fig. Sample size and matching summary.**
(DOCX)

**S5 Table. Supplementary Results.**
(DOCX)

**S1 File. Supplementary material-Data.**
(XLSX)

## Acknowledgments

The authors wish to thank all the study participants and health facilities for participating in the iHOPE study. We are very grateful to all the field workers who helped in the data collection. We also appreciate the technical and logistical support from the staff of Navrongo Health Research Centre and INDEPTH-network.

## Author contributions

**Conceptualization:** Isaiah Awintuen Agorinya, Amanda Ross, Gabriela Flores, James Akazili, Tessa Tan-torres Edejer, Kim van Wilgenburg, Maxwell Ayindenaba Dalaba, Nathan Kumasenu Mensah, Le My Lan, Yadeta Dassie Bacha, Abraham Rexford Oduro, Fabrizio Tediosi.

**Data curation:** Isaiah Awintuen Agorinya, Amanda Ross, Gabriela Flores, James Akazili, Tessa Tan-torres Edejer, Kim van Wilgenburg, Maxwell Ayindenaba Dalaba, Nathan Kumasenu Mensah, Le My Lan, Yadeta Dassie Bacha, Jemima Sumboh.

**Formal analysis:** Isaiah Awintuen Agorinya, Amanda Ross, Gabriela Flores, Maxwell Ayindenaba Dalaba, Le My Lan, Yadeta Dassie Bacha.

**Funding acquisition:** Gabriela Flores, James Akazili, Tessa Tan-torres Edejer, Fabrizio Tediosi.

**Investigation:** Isaiah Awintuen Agorinya, Amanda Ross, Gabriela Flores, James Akazili, Tessa Tan-torres Edejer, Kim van Wilgenburg, Maxwell Ayindenaba Dalaba, Le My Lan, Yadeta Dassie Bacha, Jemima Sumboh, Fabrizio Tediosi.

**Methodology:** Isaiah Awintuen Agorinya, Amanda Ross, Gabriela Flores, James Akazili, Tessa Tan-torres Edejer, Kim van Wilgenburg, Nathan Kumasenu Mensah, Le My Lan, Yadeta Dassie Bacha, Jemima Sumboh, Fabrizio Tediosi.

**Project administration:** Isaiah Awintuen Agorinya, Amanda Ross, Gabriela Flores, James Akazili, Tessa Tan-torres Edejer, Maxwell Ayindenaba Dalaba, Fabrizio Tediosi.

**Resources:** Tessa Tan-torres Edejer.

**Supervision:** Amanda Ross, Gabriela Flores, James Akazili, Tessa Tan-torres Edejer, Maxwell Ayindenaba Dalaba, Nathan Kumasenu Mensah, Yadeta Dassie Bacha, Fabrizio Tediosi.

**Writing – original draft:** Isaiah Awintuen Agorinya, Amanda Ross, Gabriela Flores, Maxwell Ayindenaba Dalaba, Nathan Kumasenu Mensah.

**Writing – review & editing:** Isaiah Awintuen Agorinya, Amanda Ross, Gabriela Flores, James Akazili, Tessa Tan-torres Edejer, Kim van Wilgenburg, Maxwell Ayindenaba Dalaba, Nathan Kumasenu Mensah, Le My Lan, Yadeta Dassie Bacha, Jemima Sumboh, Abraham Rexford Oduro, Fabrizio Tediosi.

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
