## [Decision Letter · Decision Letter 0]

13 Oct 2023

PONE-D-23-23504The effect of recall period on reported out-of-pocket health expenditure in GhanaPLOS ONE

Dear Dr. Agorinya,

Thank you for submitting your manuscript to PLOS ONE. After careful consideration, we feel that it has merit but does not fully meet PLOS ONE’s publication criteria as it currently stands. Therefore, we invite you to submit a revised version of the manuscript that addresses the points raised during the review process.

We look forward to receiving your revised manuscript.

Kind regards,

Denny John

Academic Editor

PLOS ONE

Journal Requirements:

Additional Editor Comments:

The authors are suggested to go through reviewer comments and submit a revised version.

Reviewers' comments:

Reviewer's Responses to Questions

**Comments to the Author**

1. Is the manuscript technically sound, and do the data support the conclusions?

Reviewer #1: Yes

Reviewer #2: Yes

2. Has the statistical analysis been performed appropriately and rigorously? 

Reviewer #1: Yes

Reviewer #2: Yes

3. Have the authors made all data underlying the findings in their manuscript fully available?

Reviewer #1: Yes

Reviewer #2: Yes

4. Is the manuscript presented in an intelligible fashion and written in standard English?

Reviewer #1: Yes

Reviewer #2: Yes

5. Review Comments to the Author

Reviewer #1: Feedback for Authors

Generally, the article seeks a worthy course but some comments must be addressed as shown below;

The effect of a recall period on reported out-of-pocket health expenditure in Ghana

Abstract

Authors should state the implications of their findings.

Background

The background of the study requires a very strong justification. Why was the study done in the study area? This should come out strongly in the background. The last paragraph of the background statement should be moved to the study setting.

Methods

The figure under randomization should be given a title and the source of it well provided. Reference should be made to the figure before introducing it.

Providers

Please, are you talking about health facilities or health providers?

Healthcare financing in the study setting

Issues about healthcare financing should be moved to the study setting.

Out-of-pocket payments in the study setting

Issues about out-of-pocket payments should either be moved to the study setting or the background information.

Results

Authors should check the labelling of the tables due to inconsistencies identified during the review.

Conclusions

Please, state how relevant the findings are or what are the implications of the study findings.

There are typographical issues that must be addressed. Therefore, proofreading the article will be very helpful.

Best wishes

Reviewer #2: Many thanks to the authors this interesting methodological paper. The research uses Ghanaian data to study the effect of recall period on validity of out-of-pocket health expenditures (OOPHEs) using data from a household surveys and healthcare providers. Though statistically insignificant, the authors report a higher OOPHE for the shorter recall period than the longer recall period. The paper also reports that household-reported OOPHEs are higher than provider records. It is well written, applies seemingly appropriate methods, and provides an excellent motivation in the introduction.

Page 3, line 80: A space required between "OOP(s)" and "accounted".

6. PLOS authors have the option to publish the peer review history of their article (what does this mean? ). If published, this will include your full peer review and any attached files.

**Do you want your identity to be public for this peer review?** For information about this choice, including consent withdrawal, please see our Privacy Policy .

Reviewer #1: No

Reviewer #2: No

---

## [Author Response · Author response to Decision Letter 1]

3 Apr 2024

Response to editor and reviewers’ comments

Below is a detailed response to the comments raised by the editor and reviewers. For clarity, the comments have been disaggregated according to the points raised by the reviewers and addressed accordingly.

Abstract

Comment: Authors should state the implications of their findings.

Response: The abstract has now been updated to reflect the implications of the findings

Background

Comment: The background of the study requires a very strong justification. Why was the study done in the study area? This should come out strongly in the background. The last paragraph of the background statement should be moved to the study setting.

Response: Justification has now up revised in the background section. The last paragraph of the background has also been revised to convey the original thoughts of the authors and also satisfy the comments of the reviewer.

Methods

Comment: The figure under randomisation should be given a title and the source of it well provided. Reference should be made to the figure before introducing it.

Response: The manuscript has now been revised accordingly

Providers

Comment: Please, are you talking about health facilities or health providers?

Response: In this context, health facilities and health providers mean the same. This has been clarified in the manuscript to avoid any confusion.

Healthcare financing in the study setting

Comment:

Issues about healthcare financing should be moved to the study setting.

Out-of-pocket payments in the study setting

Issues about out-of-pocket payments should either be moved to the study setting or the background information.

Response: The section has been revised accordingly

Results

Comment: Authors should check the labelling of the tables due to inconsistencies identified during the review.

Response: Tables have now been appropriately labelled.

Conclusions

Comment: Please, state how relevant the findings are or what are the implications of the study findings.

Response: The conclusion has now been revised to include the implications of the study findings.

Comment: There are typographical issues that must be addressed. Therefore, proofreading the article will be very helpful.

Response: The entire manuscript has been proofread to address typographical issues

---

## [Decision Letter · Decision Letter 1]

17 Sep 2024

PONE-D-23-23504R1The effect of recall period on reported out-of-pocket health expenditure in GhanaPLOS ONE

Dear Dr. Agorinya,

Thank you for submitting your manuscript to PLOS ONE. After careful consideration, we feel that it has merit but does not fully meet PLOS ONE’s publication criteria as it currently stands. Therefore, we invite you to submit a revised version of the manuscript that addresses the points raised during the review process.

We look forward to receiving your revised manuscript.

Kind regards,

Ugochukwu Anthony Eze

Academic Editor

PLOS ONE

Journal Requirements:

Additional Editor Comments:

Kindly address the queries of the 1st author. Minor Revision

Reviewers' comments:

Reviewer's Responses to Questions

**Comments to the Author**

1. If the authors have adequately addressed your comments raised in a previous round of review and you feel that this manuscript is now acceptable for publication, you may indicate that here to bypass the “Comments to the Author” section, enter your conflict of interest statement in the “Confidential to Editor” section, and submit your "Accept" recommendation.

Reviewer #1: All comments have been addressed

Reviewer #2: (No Response)

Reviewer #3: (No Response)

2. Is the manuscript technically sound, and do the data support the conclusions?

Reviewer #1: Yes

Reviewer #2: Yes

Reviewer #3: Yes

3. Has the statistical analysis been performed appropriately and rigorously? 

Reviewer #1: Yes

Reviewer #2: Yes

Reviewer #3: Yes

4. Have the authors made all data underlying the findings in their manuscript fully available?

Reviewer #1: Yes

Reviewer #2: Yes

Reviewer #3: Yes

5. Is the manuscript presented in an intelligible fashion and written in standard English?

Reviewer #1: Yes

Reviewer #2: Yes

Reviewer #3: Yes

6. Review Comments to the Author

Reviewer #1: Please, the authors have addressed all my comments. The paper is well written but a final proofreading before publication will be helpful.

Reviewer #2: I recommended acceptance in my earlier submission, so there's nothing more to add. The paper is well-written and accomplishes its objectives.

Reviewer #3: This is a very good study, important at this point in time in Africa in order to achieve universal health coverage. In addition the study brings to light some burden of out of pocket payment for healthcare with regards to adequate data collection, if to be used as secondary data. The claims by the authors and the significance of this study are therefore more than reasonable and justifiable. More so your data and analysis fully supports you peoples' claims in this study. Still there are some comments from me;

First, I guess the final sample size you used for the study was not less than the minimum calculated sample size for the study? How was matching of OOPs to provider records done at the individual level? What was the study duration? In your result section, there are ? two tables 3. I recommend you add percentages to the first table 3 and do not start a sentence with of.

Secondly, add more to your discussion section. You have enough results in the study to make the discussion more robust. Example is in lines 404 and 405, that states..............For these reasons, the provider data is not considered as a validation of the household-reported. You can further support this by adding that except there are proper documentation and that are verifiable. Also look at it more from the area/angle of universal health coverage and public health relevance and implications.

Thirdly, with regards to your ethical considerations, there should be feedback to the health facilities that were part of the study, the patients and their households on the findings of the study. Thank you for reaching out to them and informing them of the proposed publication of findings from the study, this ethically carries the community along. Fourthly, the list of abbreviations, should be written alphabetically, except the journal specifies otherwise and I do not think there is place for abbreviations such as &. instead use and, spelt in full.

Also, your list of references needs to be well reviewed, see what the journal recommends. Though I do not think you should just reference World Health Organization as WHO in your list of references. There are other references with with minor mistakes, go through and make the necessary corrections.

7. PLOS authors have the option to publish the peer review history of their article (what does this mean? ). If published, this will include your full peer review and any attached files.

**Do you want your identity to be public for this peer review?** For information about this choice, including consent withdrawal, please see our Privacy Policy .

Reviewer #1: No

Reviewer #2: No

Reviewer #3: **Yes: ** DR. ADESUWA QUEEN AIGBOKHAODE

---

## [Author Response · Author response to Decision Letter 2]

3 Apr 2025

Response to editor and reviewers’ comments

Below is a detailed response to the comments raised by the third reviewer. For clarity, the comments have been disaggregated according to the points raised by the reviewer and addressed accordingly.

Reviewer comment: First, I guess the final sample size you used for the study was not less than the minimum calculated sample size for the study? How was matching of OOPs to provider records done at the individual level? What was the study duration? In your result section, there are? two tables 3. I recommend you add percentages to the first table 3 and do not start a sentence with of.

Author response: There are no two (2) tables labelled 3, the table labelled 3 in the manuscript already has appropriate percentages provided (see line 329).

Reviewer comment: Secondly, add more to your discussion section. You have enough results in the study to make the discussion more robust. Example is in lines 404 and 405, that states..............For these reasons, the provider data is not considered as a validation of the household-reported. You can further support this by adding that except there are proper documentation and that are verifiable. Also look at it more from the area/angle of universal health coverage and public health relevance and implications.

Author response: This has now been addressed in the manuscript, see line 405 - 407

Reviewer comment: Thirdly, with regards to your ethical considerations, there should be feedback to the health facilities that were part of the study, the patients and their households on the findings of the study. Thank you for reaching out to them and informing them of the proposed publication of findings from the study, this ethically carries the community along.

Author response: Feedback to community and facilities was done during the dissemination of findings in the iHOPE project.

Reviewer comment: Fourthly, the list of abbreviations, should be written alphabetically, except the journal specifies otherwise and I do not think there is place for abbreviations such as &. instead use and, spelt in full.

Also, your list of references needs to be well reviewed, see what the journal recommends. Though I do not think you should just reference World Health Organization as WHO in your list of references. There are other references with with minor mistakes, go through and make the necessary corrections.

Author response: The list of abbreviations has now been arranged alphabetically (See line 446 – 476). Other minor corrections per the reviewer comments have also been effected in the manuscript.

---

## [Editor Report · Decision Letter 2]

11 Apr 2025

PONE-D-23-23504R2The effect of recall period on reported out-of-pocket health expenditure in GhanaPLOS ONE

Dear Dr. Agorinya,

Thank you for submitting your manuscript to PLOS ONE. After careful consideration, we feel that it has merit but does not fully meet PLOS ONE’s publication criteria as it currently stands. Therefore, we invite you to submit a revised version of the manuscript that addresses the points raised during the review process.

We look forward to receiving your revised manuscript.

Kind regards,

Ugochukwu Anthony Eze

Academic Editor

PLOS ONE

Journal Requirements:

Additional Editor Comments:

This is a very good work. However the discussion is very scanty. If the authors believe they have exhausted the discussion points, they should at least make som SMART recommendations on how the limitation or challenges of this study can be mitigated in future

---

## [Author Response · Author response to Decision Letter 3]

25 Sep 2025

Response to editor and reviewers’ comments

Below is a detailed response to the comments raised by the editor and reviewers. For clarity, the comments have been disaggregated according to the points raised by the reviewers and addressed accordingly.

Abstract

Comment: Authors should state the implications of their findings.

Response: The abstract has now been updated to reflect the implications of the findings

Background

Comment: The background of the study requires a very strong justification. Why was the study done in the study area? This should come out strongly in the background. The last paragraph of the background statement should be moved to the study setting.

Response: Justification has now up revised in the background section. The last paragraph of the background has also been revised to convey the original thoughts of the authors and also satisfy the comments of the reviewer.

Methods

Comment: The figure under randomisation should be given a title and the source of it well provided. Reference should be made to the figure before introducing it.

Response: The manuscript has now been revised accordingly

Providers

Comment: Please, are you talking about health facilities or health providers?

Response: In this context, health facilities and health providers mean the same. This has been clarified in the manuscript to avoid any confusion.

Healthcare financing in the study setting

Comment:

Issues about healthcare financing should be moved to the study setting.

Out-of-pocket payments in the study setting

Issues about out-of-pocket payments should either be moved to the study setting or the background information.

Response: The section has been revised accordingly

Results

Comment: Authors should check the labelling of the tables due to inconsistencies identified during the review.

Response: Tables have now been appropriately labelled.

Conclusions

Comment: Please, state how relevant the findings are or what are the implications of the study findings.

Response: The conclusion has now been revised to include the implications of the study findings.

Comment: There are typographical issues that must be addressed. Therefore, proofreading the article will be very helpful.

Response: The entire manuscript has been proofread to address typographical issues

---

## [Editor Report · Decision Letter 3]

18 Nov 2025

The effect of recall period on reported out-of-pocket health expenditure in Ghana

PONE-D-23-23504R3

Dear Dr. Agorinya,

We’re pleased to inform you that your manuscript has been judged scientifically suitable for publication and will be formally accepted for publication once it meets all outstanding technical requirements.

Kind regards,

Ugochukwu Anthony Eze

Academic Editor

PLOS ONE
---

## [Editor Report · Acceptance letter]

PONE-D-23-23504R3

PLOS One

Dear Dr. Agorinya,

I'm pleased to inform you that your manuscript has been deemed suitable for publication in PLOS One. Congratulations! Your manuscript is now being handed over to our production team.

Kind regards,

on behalf of

Dr. Ugochukwu Anthony Eze

Academic Editor

PLOS One